# Cardiac T-Tubule cBIN1-Microdomain, a Diagnostic Marker and Therapeutic Target of Heart Failure

**DOI:** 10.3390/ijms22052299

**Published:** 2021-02-25

**Authors:** Jing Li, Bradley Richmond, TingTing Hong

**Affiliations:** 1Department of Pharmacology and Toxicology, College of Pharmacy, University of Utah, Salt Lake City, UT 84112, USA; Jing.Li@pharm.utah.edu (J.L.); u0840583@utah.edu (B.R.); 2Nora Eccles Harrison Cardiovascular Research and Training Institute, University of Utah, Salt Lake City, UT 84112, USA; 3Diabetes and Metabolism Research Center, University of Utah, Salt Lake City, UT 84112, USA

**Keywords:** T-tubules, cBIN1, heart failure, calcium handling, biomarker, gene therapy

## Abstract

Since its first identification as a cardiac transverse tubule (t-tubule) protein, followed by the cloning of the cardiac isoform responsible for t-tubule membrane microdomain formation, cardiac bridging integrator 1 (cBIN1) and its organized microdomains have emerged as a key mechanism in maintaining normal beat-to-beat heart contraction and relaxation. The abnormal remodeling of cBIN1-microdomains occurs in stressed and diseased cardiomyocytes, contributing to the pathophysiology of heart failure. Due to the homeostatic turnover of t-tubule cBIN1-microdomains via microvesicle release into the peripheral circulation, plasma cBIN1 can be assayed as a liquid biopsy of cardiomyocyte health. A new blood test cBIN1 score (CS) has been developed as a dimensionless inverse index derived from plasma cBIN1 concentration with a diagnostic and prognostic power for clinical outcomes in stable ambulatory patients with heart failure with reduced or preserved ejection fraction (HFrEF or HFpEF). Recent evidence further indicates that exogenous cBIN1 introduced by adeno-associated virus 9-based gene therapy can rescue cardiac contraction and relaxation in failing hearts. The therapeutic potential of cBIN1 gene therapy is enormous given its ability to rescue cardiac inotropy and provide lusitropic protection in the meantime. These unprecedented capabilities of cBIN1 gene therapy are shifting the current paradigm of therapy development for heart failure, particularly HFpEF.

## 1. Introduction

Cardiomyocyte transverse tubule (t-tubule) microdomains organized by the membrane scaffolding protein cardiac bridging integrator 1 (cBIN1) have recently emerged as a master regulator of beat-to-beat myocardial contraction and relaxation. Not only are cBIN1-microdomains critical to normal cardiomyocyte biology and heart physiology, they are also remodeling in diseased cardiomyocytes. Abnormal remodeling in cBIN1-microdomains is a common pathophysiology of acquired heart failure. In this review, we will summarize the current knowledge of t-tubule cBIN1-microdomains regarding their formation, function, regulation, and turnover, as well as their potential as heart failure biomarkers and therapeutic targets.

## 2. Basic Biology and Pathophysiology of cBIN1-Microdomains in Normal and Failing Hearts

### 2.1. The Cardiac T-Tubule System and Its Microdomains

Cardiac t-tubules are sarcolemmal invaginations which form a dynamic membrane system important for excitation–contraction (EC) coupling via its role in organizing ion channels and other membrane proteins critical to the intracellular process of calcium-induced calcium release (CICR). In cardiomyocytes, t-tubules form a complex network comprising approximately 21 to 64% of the total sarcolemma (see review in [1]). Larger and more heterogenous than skeletal muscle t-tubules [1], cardiac t-tubules are also highly dynamic, allowing for the efficient regulation of EC coupling and membrane excitability, particularly during the acute stress response. While little is known about the mechanisms of t-tubule biogenesis or dedifferentiation, it is well accepted that mature and healthy t-tubules are labile and exhibit a constant turnover of membrane, ion channels, and membrane proteins [1]. To accomplish this turnover and to ensure pools of easily accessible channel proteins during stress, t-tubule components tightly regulate processes such as the forward trafficking, recycling, and degradation of ion channels and other proteins [1]. Many of the channels common to the t-tubules contain multiple signaling sequences that regulate their synthesis and trafficking [2,3,4], while their short functional half-lives compared to their longer total half-lives indicate the use of recycling pathways [5,6,7,8]. During sympathetic stimulation or other forms of stress, these regulatory pathways allow t-tubules to rapidly change their lipid and protein compositions to maintain efficient EC coupling and electrical stability [9].

Cardiac t-tubules accomplish this complex system of regulation via well-organized membrane microdomains which are normally created by membrane scaffolding proteins. Different microdomains have their own pools of ion channels, transporters, and signaling molecules to accomplish unique functions. For instance, caveolae are cholesterol and sphingomyelin-rich microdomains organized by the caveolin-3 scaffolding protein which house a subpopulation of L-type calcium channels (LTCCs) together with β2 adrenergic receptors, facilitating channel modulation during sympathetic stress [10]. Ankyrin-B microdomains, on the other hand, are enriched with the sodium calcium exchanger/sodium potassium ATPase/inositol trisphosphate receptor complex responsible for both ensuring calcium decline during relaxation and extruding excess calcium that might interfere with CICR efficiency [11]. The mutation or disruption of either the ankyrin-B or caveolin-3 scaffolding proteins can result in conditions such as sick sinus syndrome, long QT syndrome, arrhythmias, and abnormal heart rates [10,12,13]. These distinct microdomains thus allow t-tubules to carry out a variety of specialized yet critical functions regulated by their own distinctive set of proteins and signaling pathways. In the following section, we will discuss in detail the cBIN1-microdomains, which are the microdomains involved in intra-cardiomyocyte calcium handling. These phospholipid-rich cBIN1-microdomains house LTCCs and initiate calcium transients during each heartbeat [14]. The function of cBIN1-microdomains, as well as the potential of targeting cBIN1 to develop new biomarkers and novel therapeutics for heart failure, is the main subject discussed in this review.

### 2.2. T-Tubule cBIN1-Microdomains

Formation—Cardiomyocyte cBIN1-microdomains are created by t-tubule membrane microfolds sculpted by a membrane curvature formation protein cBIN1. BIN1 (amphiphysin 2) is an N-terminal Bin1-Amphiphysin-Rvs (N-BAR) domain-containing protein that is widely expressed [15]. BIN1 was first identified as a tumor suppressor due to its interaction with the MYC oncoprotein [16]. Encoded by one gene with 20 exons, over ten BIN1 protein isoforms are detected across a variety of tissues. Tissues from the brain, heart, and skeletal muscle express ubiquitous and tissue-specific BIN1 isoforms via alternative splicing [17,18,19,20]. The defining constitutive N-BAR domain, encoded by exons 1–10 at the N-terminus, is responsible for BIN1 dimerization and the formation of its characteristic “banana” shape [21]. The constitutive exons 19–20 encode a C-terminus SH3 domain that is crucial for interactions with other proteins and cytoskeletal elements [22,23]. The MYC-binding domain is formed by a ubiquitously alternatively spliced exon 17 and a constitutive exon 18 [24]. Exons in the middle region undergo heavy alternative splicing with tissue and disease specificities. For instance, isoforms containing an exon 11-encoded phosphoinositide-binding motif are highly expressed in muscle [25], promoting membrane tubulation during skeletal t-tubule biogenesis. On the other hand, the neuronal system co-splices exons 13–16 together to encode a clathrin-associated protein (CLAP) binding region which is important for endocytosis and neurotransmitter reuptake [18,19].

In the cardiac system, the alternate splicing of *Bin1* produces 4–6 transcript variants and the corresponding protein isoforms. The two cardiac characteristic isoforms feature the inclusion of exon 13 (deficient of exons 14–16) with or without the ubiquitously alternatively spliced exon 17 to form the protein isoforms of BIN1+13 and BIN1+13+17 (now called cBIN1) [17,20]. Of these two isoforms, cBIN1 is the isoform which localizes to the cardiac t-tubules and there acts as the critical membrane scaffolding protein that is the focus of this review [17]. Previous work identified that co-splicing with exons 13 and 17 in cBIN1 is needed for the formation of cardiac t-tubule microfolds and the effective organization of microdomains critical to normal intracellular calcium handling [17]. By interacting with both the lipid bilayer and subsarcolemmal cortical actin, cBIN1 maintains these membrane microdomains in homeostasis at the t-tubule network. The composition and function of the cBIN1-microdomains will be discussed in detail in the next section. Interestingly, exon 11-containing skeletal BIN1 isoforms absent from mouse cardiomyocytes are expressed in sheep hearts [26], indicating a potential role of skeletal BIN1 in cardiac t-tubule biology. Furthermore, the overexpression of the exon 11-containing skeletal BIN1 isoform 8 can induce narrow tubular structures with retained calcium channel trafficking capacity in HL-1 cells [27], human embryonic stem cell-derived cardiomyocytes [28], neonatal rat ventricular myocytes, and iCell induced pluripotent stem cell (iPSC)-derived cardiac myocytes [26]. Future studies will be necessary to understand the expression profiles of BIN1 isoforms and their functional significance during myocyte maturation in humans and other species of large mammals.

Function—cBIN1-microdomains at cardiac t-tubules facilitate efficient EC coupling by organizing dyads formed by voltage-gated LTCCs and ryanodine receptors (RyRs) (Figure 1A). T-tubule cBIN1 facilitates the microtubule-dependent forward trafficking of LTCCs, resulting in LTCC enrichment at t-tubules accounting for 80% of overall cell surface LTCCs [27,29]. Once delivered and expressed at the t-tubule surface, LTCCs are also clustered to cBIN1-microdomains, which recruit RyRs at the junctional sarcoplasmic reticulum (jSR) for effective couplon formation with LTCC clusters at t-tubules [14,17,30]. By bringing RyRs and LTCCs into proximity, cBIN1-microdomains help organize the calcium-releasing unit dyads where the normal calcium transient develops for efficient EC coupling. In addition to systolic calcium release, cBIN1-microdomains were recently found to be critical for diastolic calcium removal via organizing the intracellular distribution of the sarco/endoplasmic reticulum Ca^2+^ ATPase (SERCA2a) along the sarcoplasmic reticulum (SR) membrane [31]. cBIN1-microdomains help maintain a subpopulation of SERCA2a near the jSR to facilitate the diastolic reuptake of calcium into the SR critical for proper heart relaxation [31]. In addition to its role in regulating intracellular calcium cycling, cBIN1 promotes electrical stability in heart muscle cells. The banana-shaped cBIN1 dimers-sculpted tortuous membrane folds restrict extracellular ion diffusion within t-tubule lumen [17], creating a slow diffusion zone similar to the intracellular “fuzzy space” [32]. During increasing rates of contraction, the slow diffusion zone causes outward-flowing ions like potassium to accumulate while inward-current ions like calcium deplete quickly, protecting cardiomyocyte electrical stability [33].

cBIN1-microdomains also contribute to the highly dynamic nature of t-tubule networks. As previously mentioned, the t-tubule network is highly susceptible to change. It undergoes constant turnover, remodels in heart failure, and breaks down when cardiomyocytes are cultured (see review in [1]). It is possible that these changes are accomplished in part by the endocytosis of cBIN1-microdomains given the known interaction between BIN1 and dynamin 2 [34]. On the other hand, it has been recently identified that cBIN1 facilitates endosomal sorting complexes required for transport (ESCRT)-dependent formation and the release of microvesicles [35], which may also be recycled as suggested in the role of neuronal BIN1 isoforms in neurotransmitter reuptake [18,35]. Given its roles in both intracellular (endocytosis) and extracellular (ectocytosis) recycling, cBIN1 is likely directly involved in the constant membrane turnover of the t-tubule network.

### 2.3. cBIN1-Microdomain Remodeling in Heart Failure

Heart failure is an end-stage cardiovascular syndrome with high morbidity and mortality. Heart failure morbidity is caused by impaired contractility and relaxation-induced pump failure, as well as sudden cardiac death due to lethal ventricular arrhythmias. For now, the best therapies available include heart transplants, left ventricular assist devices (LVADs), and implantable cardioverter defibrillators (ICDs). These treatments are often expensive, difficult to procure, or are implanted wastefully [36,37,38,39,40,41]. Because of the limited knowledge available about cardiomyocyte health and recovery, it is difficult to correctly prioritize hearts for current treatments or develop new cell-based therapies. For this reason, it is important to study and identify the molecular changes that accompany heart failure. One molecular change of note is the reduction in cBIN1 transcription in acquired human heart failure [29] and other animal models of heart failure [42,43]. While the cause of this reduction remains unclear, results from recent studies indicate that cBIN1 expression is reduced when the myocardium is subjected to common stressors such as sympathetic overdrive induced by chronic isoproterenol infusion [31] and continuous pressure overload induced by transverse aortic constriction [44]. Importantly, reductions in cBIN1 have important ramifications for the health of cardiomyocytes in failing hearts. Indeed, a transcriptional reduction in cBIN1 levels is a common characteristic of acquired human heart failure [29], indicating the biomarker potential of cBIN1. The mechanisms and clinical development of cBIN1 as a marker of cardiac muscle health will be discussed in Section 3 of the current review.

Given the many ways that cBIN1 facilitates healthy cardiomyocyte physiology, it is unsurprising that myocardial function worsens as cBIN1 transcriptional levels drop. The first evidence of cBIN1 as an indispensable part of normal cardiac t-tubule function and myocardial performance was found in constitutive *Bin1* knockout mice, which develop perinatal lethality due to cardiomyopathy. Follow-up studies proved that the cardiac-specific ablation of the *Bin1* gene leads to dilated cardiomyopathy with reduced ejection fraction in aged or stressed mice [31,45]. Notably, though cBIN1 only accounts for a small percentage (8%) of all BIN1 isoforms in mouse hearts, it plays a critical role in t-tubule membrane microdomain formation [17,29]. Pathophysiological changes caused by reduced cBIN1 in failing cardiomyocytes include: (1) impaired microtubule-dependent LTCC forward trafficking, resulting in the reduced expression of LTCCs at the t-tubule surface [27]; (2) altered LTCC clustering with impaired channel function [27]; (3) removal of the protective slow ionic diffusion zone due to t-tubule microfold loss, leading to susceptibility to ventricular arrhythmia [17,32]; (4) inefficient RyR recruitment to the jSR leading to fewer LTCC–RyR dyads and an accumulation of orphaned leaky RyRs [14,27,30]; and (5) disorganized SERCA2a at the SR which causes impaired calcium removal and diastolic dysfunction [31] (Figure 1B). Most excitingly, our recent work indicates that the pathophysiology associated with cBIN1-microdomain remodeling in failing hearts can be normalized by exogenous cBIN1 introduced by gene therapy. This rescue capacity of exogenous cBIN1 further indicates that cBIN1 is a master regulator of the calcium handling machinery in cardiomyocytes and that the disruption of cBIN1-microdomains is a cause of calcium mishandling in failing hearts. The potential of *cBIN1* gene therapy for heart failure will be reviewed in detail in Section 4 of the current review. Future studies are also needed to understand whether alterations in intra-myocyte t-tubule cBIN1-microdomains influence heart failure development promoted by factors outside of cardiomyocyte pathophysiology such as inflammation, fibrosis, and remodeling in extracellular matrix.

## 3. cBIN1 as a Marker of Heart Failure—Basic Mechanisms and Clinical Observations

### 3.1. Current Status of Heart Failure Biomarkers

Heart failure is a syndrome that is difficult to manage. For advanced heart failure, the current guidelines for ICD and LVAD implantation are based on clinical criteria including ejection fraction and New York Heart Association (NYHA) classification, both of which are subjective and imperfect measures of outcome. Current AHA guidelines suggest that the use of biomarkers may be helpful for risk stratification [46]. However, none of the currently available heart failure biomarkers have the capability to identify the health and recovery potential of cardiac muscle. Available heart failure biomarkers [40] include non-specific markers of inflammatory response (C-reactive protein [47], pro-inflammatory cytokines [48]), extracellular matrix remodeling (matrix metalloproteinases) [49], cardiac loading and atrial stretch (B-type natriuretic peptide, BNP family [50]), whole body neurohormonal activation (suppression of tumorigenicity 2, ST2) [51], and inflammation (galectin-3) [52]. Unfortunately, these biomarkers are not specific to ventricular muscle and can be affected by age, obesity, systemic inflammation, and other common co-morbidities such as renal failure and pulmonary hypertension. The family of natriuretic peptides (BNP and N-terminal pro-hormone BNP, NT-proBNP) is excellent in the setting of attributing dyspnea of unclear origin to cardiogenic fluid overload [53]. However, natriuretic peptides do not help diagnose asymptomatic patients and as established by the recent Guiding Evidence Based Therapy Using Biomarker Intensified Treatment (GUIDE-IT) trial, natriuretic peptides which assess fluid status do not enhance the standard of care management of chronic heart failure [54]. Due to these limitations, novel biomarkers that measure cellular cardiac health and recovery potential in heart failure are much needed to guide medical therapy.

Meanwhile, utilizing the cardiac specificity of cBIN1 as a biomarker of muscle health rather than inflammation or loading conditions is a new and substantive departure from the status quo. Recent insights into t-tubule membrane turnover mechanisms and pathways have revealed that the integrity of cardiomyocyte t-tubule cBIN1-microdomains can be assayed using a blood test to serve as a marker of the biochemical health of cardiac muscle cells. Furthermore, the decrease in cBIN1 with systolic or diastolic stress [29,30,31,43] and cBIN1 blood availability [35,55] provide the opportunity for its use by clinicians to quantitatively index the extent of cardiac muscle failure in their heart failure patients. In this section, we will discuss the mechanisms underlying the blood availability of cBIN1 and recent clinical studies of cBIN1 that evaluate its clinical usage as a biomarker facilitating heart failure diagnosis and prognosis.

### 3.2. Blood Availability of Cardiac cBIN1—Turnover of cBIN1-Microdomains as Microvesicles

Biogenesis of cBIN1-microvesicles—cBIN1-organized membrane microdomains at t-tubules are extremely dynamic. On one hand, the C-terminal constitutive SH3 domain of BIN1 binds to dynamin 2 and regulates dynamin 2-dependent endocytosis [56]. It is likely that dynamin-2 is involved in intracellular recycling of t-tubule cBIN1-microfolds through the formation of endocytic vesicles. The inhibition of dynamin-2 by dynasore may help stabilize cardiomyocyte t-tubule membrane, protecting cardiac function as previously reported [57]. On the other hand, the N-terminal BAR domain in BIN1 molecules dimerizes to form a banana-shaped topology, which not only binds to the lipid bilayer but also recruits protein members in the ESCRT-III family to initiate ectocytosis at the t-tubule membrane [35]. Recruitment of the ESCRT-III family member charged multivesicular body protein 4b (CHMP4B) to the cBIN1 membrane facilitates microparticle formation and release from t-tubules to the extracellular environment (Figure 2). The binding affinity between N-BAR in BIN1 and CHMP4B is regulated by the actin cytoskeleton [35], which promotes vesicle release when strengthened. BIN1′s interaction with CHMP4B requires its constitutive N-terminal BAR domain [35], which is present in all BIN1 isoforms. Interestingly, BIN1+17 and cBIN1 are the only two BIN1 isoforms circulating in human and mouse plasma despite the higher expression levels of other BIN1 isoforms in heart muscle. Similarly, the exon 11-contianing BIN1 isoform 8 highly expressed in skeletal muscle is also not detectable in the peripheral blood stream [35]. These data indicate that additional regulation and protein interaction are required for the cellular release of BIN1 containing vesicles. Although the cell origin and functional significance of plasma BIN1+17 vesicles remain unclear, a comparison between the nonspecific assay detecting BIN1+17 and cBIN1 in human plasma [55] and the cBIN1 specific test [58,59] indicates a lack of correlation between plasma BIN1+17 and cardiac function. Future studies will be necessary to understand the BIN1 isoform-specific regulation of microvesicle release from cells.

The roles of cBIN1 in promoting microparticle release and anchoring t-tubule membranes to Z-discs are both actin-regulated cellular processes needed to maintain an intracellular homeostatic equilibrium of the calcium regulatory cBIN1-microdomains. In stressed or diseased cardiomyocytes, abnormal intracellular formation and/or the extracellular recycling of cBIN1-microdomains may disrupt the normal physiological equilibrium and cause a shift towards a new equilibrium with fewer homeostatic cBIN1-microfolds, thus reducing the calcium regulatory capacity of t-tubules. As a result of reduced cBIN1, diseased cardiomyocytes cannot efficiently mobilize calcium handling machinery during acute stress, diminishing the necessary responsive gain in EC coupling and myocardial functional reserve. To further understand the cellular processes involved in the maintenance of cBIN1-microdomains and the biogenesis of cBIN1-vesicles, future studies will be necessary to explore the interplay among the membrane bending and scaffolding capacities of BIN1 proteins, the ESCRT family members, the cytoskeleton, the intracellular calcium kinetics, and calcium-dependent enzymes.

Content and fate of extracellular cBIN1-vesicles—While it is known that cBIN1-vesicles released from cardiac t-tubule membrane carry the signature cBIN1 protein [35], other molecular cargoes contained in cardiac origin cBIN1-vesicles await future exploration. It is possible that circulating cBIN1-vesicles enclose other characteristic components of cardiomyocyte origin such as the proteins and regulatory molecules of the calcium handling machinery. When reabsorbed to cardiomyocytes, these components could play a role in regulating intracellular calcium cycling in the recipient cells.

Interestingly, turnover of BIN1-membrane through extracellular vesicle release is not unique to cardiomyocytes. Indeed, extracellular vesicle release from BIN1-containing cell membranes is observed in neurons and microglial cells in the central neural system (CNS) [60]. In addition to BIN1 molecules, these extracellular BIN1-vesicles of CNS origin contain other molecular components. For instance, in Alzheimer’s disease, the BIN1-vesicles of microglial origin contain tau protein and have paracrine seeding capacity on neurons, thus serving as a tau spreading mechanism during Alzheimer’s disease progression [60].

In contrast to the CNS BIN1-vesicles, the cardiac-released cBIN1-vesicles are enriched in the peripheral blood stream (Figure 2) [35]. These circulating cBIN1-vesicles may simply serve as a protective cellular disposal mechanism for cardiomyocytes, but the exploration of molecular cargoes within the circulating cBIN1-vesicles could indicate otherwise. Whether these cBIN1-vesicles of cardiac origin contain functional molecules or have significant autocrine or paracrine function are topics that await future studies. For instance, it is possible that these vesicles may beneficially influence the local extracellular environment of the heart, that these vesicles when reabsorbed may have differential effects on vesicle-releasing versus non-releasing cardiomyocytes, and that the cardiac released vesicles may also travel far in the peripheral circulation to generate a remote impact on other organ systems such as skeletal muscle. Further in-depth research will be critical to understand the content, destination, and function of these cBIN1-vesicles released from cardiomyocyte t-tubule microdomains. These studies may help reveal new molecular pathways that can be targeted for the development of new therapies aiming to normalize cBIN1-microdomains in failing cardiomyocytes.

### 3.3. Clinical Utility of the Plasma cBIN1–Vesicles Derived CS Test

CS, a new blood test for heart failure—Given their significance in sustaining optimal intra-myocyte beat-to-beat calcium transients, cBIN1-microdomains at t-tubules are emerging as a crucial regulator of myocardial function as well as a promoter of heart failure progression when disrupted. In heart failure, both stress-induced transcriptional reduction and abnormal splicing cause reduced cBIN1 protein expression, disrupting cBIN1 microdomains and promoting disease progression. Furthermore, due to the homeostatic release of extracellular cBIN1-microvesicles, the amount of cardiac t-tubule cBIN1-microdomains correlates to the level of cBIN1-vesicles in circulation. As a result, plasma cBIN1 levels are direct indicators of the amount of myocardial t-tubule cBIN1-microdomains, reflecting a stable functional reserve of cardiomyocytes [35]. Thus, quantifying blood cBIN1 can accurately measure homeostatic cardiac t-tubule cBIN1 in patients with normal or diseased hearts. In other words, plasma cBIN1 is a liquid biopsy of cardiac tissue cBIN1-microdomains.

A cBIN1-specific blood test can provide an accurate measurement of myocardial functional reserve in symptomatically stable patients, and thus provide a potential biomarker of myocardial health and reserve. A cBIN1 blood test has already been developed and explored for clinical usage in aiding heart failure management. In a previous study using a clinical cohort of arrhythmogenic right ventricular cardiomyopathy, low plasma BIN1 measured by an ELISA test identified patients developing symptomatic heart failure and predicted future arrhythmia incidences [55]. However, the study was conducted prior to the cloning of cBIN1 in cardiomyocytes and the development of the cBIN1 specific test. At the time, the utilized non-specific BIN1 blood test captured both BIN1+17 and cBIN1 in human plasma, thus explaining the large variations observed in measured plasma BIN1 concentrations in individuals from the same study cohort [55]. The identification and cloning of the microdomain-forming cBIN1 [17] has allowed the development of a cardiac-specific cBIN1 blood test using exon-specific antibodies raised against epitopes within the exons 13 and 17-encoded domains in cBIN1. As a result, it is possible to differentially measure cBIN1 concentration in human plasma with sufficient specificity and sensitivity. When combined with the osmotic shock technique used to release all cBIN1 molecules contained within plasma microvesicles [35], this cBIN1 specific ELISA test can accurately measure the plasma levels of cBIN1-vesicles in human patients to aid in the diagnosis and prognosis of patients with failing hearts [58,59].

This newly developed cBIN1 ELISA assay has been successfully used to obtain plasma cBIN1 concentrations from hundreds of healthy control patients without heart failure. These original data indicate that the log transformation of plasma cBIN1 levels in a healthy population follows a normal distribution [58]. To simplify the test for convenient clinical usage, our clinical team has developed a dimensionless inverse index of plasma cBIN1 levels normalized to the median cBIN1 level in the healthy population. This uniform scoring system has now been standardized for cBIN1 score (CS) measurement in clinical patients across study sites and clinical trials. Since CS is the inverse index of cBIN1 concentration, a lower plasma cBIN1 level in patients with heart failure corresponds to a rise in CS.

CS performance in heart failure diagnosis and prognosis—The performance of CS in facilitating heart failure diagnosis and prognosis was tested in two clinical cohorts of heart failure including patients with heart failure with reduced (HFrEF) [59] and preserved ejection fraction (HFpEF) [58]. In both studies, CS was obtained in ambulatory patients during regular clinical visits. In each cohort, the diagnostic and prognostic performance of CS was then compared to the performance of NT-proBNP. These patients were followed up for 18 months for clinical outcomes including death, cardiovascular events, and hospitalization. In these functionally stable patients, NT-proBNP successfully identified patients with heart failure, but failed to accurately predict future clinical outcomes. The lack of prognostic value of NT-proBNP is consistent with the failed results of the recent GUIDE-IT clinical trial [54], which is likely due to the intrinsic mechanism of NT-proBNP limited to its identification of volume overload and fluid retention rather than muscle heath. In contrast, as a liquid biopsy of cardiomyocyte t-tubule cBIN1-microdomains, CS accurately detects the biochemical health of cardiomyocytes, identifying the functional reserve and recovery potential of the failing myocardium. Thus, CS provides a unique opportunity to accurately measure myocyte health with unprecedent prognostic value, as suggested in the studied HFrEF and HFpEF patients. Furthermore, by respectively measuring fluid status and myocyte health, NT-proBNP and CS could be used in conjunction to measure two important and distinct pathophysiological aspects of heart failure. Combined NT-proBNP and CS tests could thus provide a sensitive and specific measurement of heart failure status. Additional clinical studies are currently evaluating the performance of CS alone or in combination with NT-proBNP in aiding heart failure diagnosis and prognosis.

Based on the data from the HFpEF cohort [58], a particular advantage of CS over other biomarkers is that it can be used to diagnose HFpEF, a form of heart failure that is difficult to detect using common diagnostic tools like left ventricular ejection fraction (LVEF) and plasma natural peptide levels. Indeed, nearly one third of HFpEF patients present with normal plasma natural peptide levels. Due to the limitations of these common diagnostic tools, HFpEF is much more difficult to diagnose than HFrEF. CS provides a crucial new diagnostic tool for accurately detecting HFpEF. Studies have already shown that CS rise is observed in HFpEF patients with normal NT-proBNP, but not in patients without heart failure from the healthy and comorbidity control cohorts. These types of studies suggest that CS is an accurate marker which can successfully distinguish failing cardiac muscle from healthy muscle regardless of overall organ function such as ejection fraction. Thus, the future classification of failing hearts can be based less on measures of overall organ function and more on the molecular details of cardiomyocyte pathophysiology.

CS and heart failure early diagnosis—More recently, we identified that cBIN1 reduction occurs in early cardiomyocyte t-tubule remodeling before myocardial functional decompensation and symptomatic heart failure [31]. Assaying t-tubule cBIN1-microdomains is therefore a possible way of detecting abnormal cardiomyocyte remodeling even in early non-symptomatic disease stages preceding heart failure. By indirectly measuring t-tubule cBIN1, the plasma cBIN1-based CS analysis is a liquid biopsy of cBIN1-microdomains in cardiomyocytes. With the recently developed CS test featuring an increased sensitivity now capable of detecting <10 ng/mL of plasma cBIN1 concentration, clinicians can detect smaller changes in t-tubule cBIN1 levels in pre-heart failure patients with cardiovascular diseases and heart failure-associated comorbidities like obesity, diabetes, and hypertension. In these pre-heart failure patients with risk factors and comorbidities, a minor to moderate rise in CS may help identify those patients who will have a higher incidence of future heart failure. Using this information, personalized clinical regimens can be strategized to limit and reduce the development of heart failure in these patients. Future longitudinal clinical studies will be helpful in evaluating the performance of CS in predicting future heart failure in patients with hypertension and diabetes.

## 4. AAV9-cBIN1 Is a Novel Gene Therapy for Heart Failure

### 4.1. Overview of Heart Failure Gene Therapy

Heart failure is a major and fast-growing public health problem affecting over 6 million Americans and 2% of the adult population worldwide [61,62]. Even with the development of new drugs and device-based therapies in recent decades [63,64], heart failure patients continue to suffer from a low quality of life and high rates of mortality due to the limited efficacy of available pharmacological treatments [63,64]. The limitations of current therapeutic approaches have led to the pursuit of new cardiac gene therapies as alternative approaches to prevent and treat heart failure. The idea of gene therapy is to directly introduce exogenous genes (transgenes) to a patient in order to restore the level of a targeted protein, in this instance, to benefit myocardial health and improve cardiac performance [65]. Insights into the cellular and molecular pathogenesis of heart failure brought to light a variety of potential therapeutic targets that are involved in cardiomyocyte function and cell signaling pathways [66,67,68,69]. While gene therapy has succeeded in many fields of clinical medicine and has yielded several approved products for clinical use, there has not yet been a gene therapy that has successfully and significantly improved clinical outcomes for heart failure [69]. Here, we will review some of the strategies and protein targets that have been explored as potential cardiac gene therapies over the years. We will also highlight the potential of cBIN1 as a novel gene therapeutic target in future studies.

The pathophysiological hallmark of a failing heart is abnormal calcium handling [70,71,72]. A healthy calcium release-and-reuptake cycle is pivotal in maintaining normal cardiac function during the contraction and relaxation phases [73,74]. Hence, targeting specific key regulators in calcium flux pathways has been the focus of most gene therapy strategies. SERCA2a is a key factor for calcium reuptake, which transports calcium from the cytosol to the SR during the diastolic phase. SERCA2a is inhibited by the de-phosphorylated form of phospholamban (PLN), while the phosphorylation of PLN relieves this inhibition. The main source of phosphorylated PLN in cardiomyocytes is the cyclic adenosine monophosphate (cAMP)-dependent protein kinase A (PKA) pathway, which is regulated by β-adrenergic stimulation [75]. Conversely, protein phosphatase 1 (PP1) de-phosphorylates PLN and leads to the inactivation of SERCA2a and impaired calcium reuptake. The PP1 de-phosphorylation activity can in turn be blocked by Inhibitor-1c (I-1c) binding to PP1 [76]. Three main gene therapy approaches targeting SERCA2a, adenylate cyclase, and I-1c have been explored and evaluated in clinical trials over the years.

SERCA2a—The first and most extensively studied strategy aimed to overexpress SERCA2a in failing hearts based on the observation that both the level and activity of SERCA2a are downregulated in human and experimental models of heart failure. Investigators succeeded in reversing cardiac dysfunction in isolated human cardiomyocytes [77], murine models [78], and porcine models [79] by SERCA2a gene transfer. After a Phase I/II trial tested the safety of intracoronary infusions of the adeno-associated virus 1 (AAV1) vector expressing SERCA2a cDNA [80], the Calcium Up-Regulation by Percutaneous Administration of Gene Therapy in Cardiac Disease (CUPID) trial was initiated as the first clinical attempt to treat heart failure with adeno-associated virus (AAV) gene therapy [81]. The CUPID trial was a Phase IIa, open-labelled, randomized, and placebo-controlled study that enrolled 39 patients with severe HFrEF. The initial trial demonstrated the safety and benefit of the AAV1/SERCA2a therapy in advanced heart failure [81], which impelled a larger confirmatory Phase IIb, double-blinded, randomized trial, CUPID2 study [82,83]. Unfortunately, the CUPID2 trial failed to improve the clinical course of patients with severe HFrEF at the studied dose of AAV1/SERCA2a viral particles. The failure of CUPID2 to yield functional benefits in recruited patients caused the premature termination of two other related Phase IIa clinical trials using the same vector: the AGENT-HF study [84] and the SERCA-LVAD study [85] in which the AAV1/SERCA2a therapy was tested in patients with left ventricular assist devices. Despite the disappointing results of CUPID2 trial, the attempt stimulated more on-going clinical trials utilizing gene therapy to treat heart failure patients.

Adenylate cyclase 6—Another approach proposed to treat heart failure involves the manipulation of β-adrenergic signaling. Adenylate cyclase 6 (AC6) catalyzes ATP to cAMP, a key factor that is generated in response to β-adrenergic receptor stimulation and activates PKA. Preclinical studies on a swine model of heart failure proved AC6 gene transfer improved cardiac function and attenuated deleterious left ventricular remodeling [86,87]. The protective effect of AC6 gene therapy was also proven in mice with genetic cardiomyopathy [88]. The promising results in animal models prompted a Phase II clinical trial in which AC6 was transduced by adenovirus 5 (Ad5.hAC6) in HFrEF patients in a randomized, double-blinded, placebo-controlled manner [89]. A one-time administration of Ad5.hAC6 gene transfer in this clinical trial safely increased left ventricular function in heart failure patients. A larger Phase III clinical trial was planned but withdrawn later because of a re-evaluation of its clinical recruitment plans and strategy [69].

I-1c—An additional alternative method of gene therapy for heart failure relies upon overexpressing the constitutively active inhibitor I-1c that binds and inhibits PP1. Inhibited PP1 leads to increased PLN phosphorylation and improved SERCA2a activity, which ultimately restores normal calcium flux and β-adrenergic stimulation. In a preclinical swine model of myocardial infarction-induced heart failure, the overexpression of I-1c by intracoronary administration of an AAV9 vector preserved cardiac function and reduced scar size [90]. A new cardiotropic vector generated by the capsid reengineering of an AAV (BNP116) carrying I-1c was further used in swine models of ischemic and nonischemic heart failure [91,92]. Both preclinical studies demonstrated that the AAV9/I-1c therapy safely improved cardiac contractility. Recently, a Phase 1, prospective, multi-center, open-label, sequential dose escalation study was initiated to explore the safety, feasibility, and efficacy of a single intracoronary infusion of BNP116.sc-CMV.I1c in patients with NYHA Class III heart failure.

Despite promising results in animal models, clinical trials of gene therapy for heart failure have not yet shown positive results. Several possible explanations for the inconsistency between the preclinical and clinical studies are: (1) low efficiency of gene delivery to cardiomyocytes: improper viral vectors, inadequate delivery methods, sub-optimal dose, etc.; (2) inappropriate animal models: inadequacy of a preclinical animal model in fully mimicking human heart failure, limited available models of heart failure in large animals; (3) wrong therapeutic target gene: insufficiency to alter failing cardiomyocytes by introducing a targeted gene [69]. Therefore, more recent strategies of gene therapy for heart failure have developed aiming for noncoding RNA therapeutic genes [93,94,95], cardiac regeneration [96,97,98,99,100], and reengineered AAVs [101,102,103]. Further studies utilizing larger cohorts of patients are needed to prove the efficacy of emerging gene therapies.

### 4.2. cBIN1: A Potential Target of Gene Therapy for Heart Failure

There is a major unmet need for effective target genes for treating heart failure. Due to the complexity of heart failure etiology and the diversity of clinical manifestations in heart failure patients, a master regulator that acts on multiple proteins and cellular pathways might be better positioned to drive the functional recovery of failing muscle. Unlike many other heart failure-associated molecules, cBIN1 creates t-tubule membrane microdomains which are involved in the multilayered regulation of calcium flux. Thus, targeting cBIN1-microdomains provides a unique opportunity to protect inotropy and lusitropy simultaneously via the effective organization of the entire calcium regulatory machinery from the systolic calcium-releasing unit dyads to the key diastolic calcium remover SERCA2a.

The therapeutic effect of cBIN1 gene therapy has been tested in two mouse models of hypertrophy and/or heart failure. The study by Liu et al. [31] investigated the effect of AAV9-mediated cBIN1 gene transfer in mice subjected to chronic isoproterenol infusion. As a synthetic catecholamine and nonselective β-adrenergic agonist, isoproterenol is widely used in research to induce left ventricular hypertrophy and diastolic dysfunction in animal models [104]. Sustained isoproterenol administration causes decreased cardiomyocyte cBIN1 levels and the disorganized intracellular distribution of calcium-handling proteins including LTCC and SERCA2a. The normalization of cBIN1 levels through AAV9-mediated gene transfer restored t-tubule organization and healthy calcium transients, improved cardiac function, and prevented heart failure progression. Interestingly, the cBIN1-treated hearts under isoproterenol challenge presented a more relaxed and hyper-efficient phenotype similar to athletic hearts [105,106,107]. Thus, cBIN1 gene therapy might bring exercise-like benefits to patients with heart failure. The observed protection of cardiac lusitropy in cBIN1-treated hearts also raises the possibility of cBIN1 as a promising new target for treating HFpEF patients, a fast-growing population who have fewer available treatment options than HFrEF patients [108,109]. The therapeutic efficacy of cBIN1 gene therapy was also confirmed in mice with transverse aortic constriction (TAC)-induced heart failure [44]. Under TAC-induced pressure overload, compensated cardiac hypertrophy progresses over time as an adaptive response, leading to irreversible cardiac dilatation and worsening HFrEF [110]. When introduced either prior to [31] or post TAC-induced heart failure [44], AAV9-mediated cBIN1 gene therapy successfully protected myocardial function and postponed or even reversed heart failure progression. These are important data establishing the rescue capacity of cBIN1 gene therapy when introduced to hearts that are already failing.

Taken together, here we have highlighted the current preclinical animal experiments that support the use of *cBIN1* gene therapy as a potential treatment for heart failure in the future. Further studies in large animals with common heart failure comorbidities will be needed before clinical trials can be pursed. Additionally, more efficient and higher cardiac-tropic methods of delivering the *cBIN1* gene will need to be explored. Overall, cBIN1 is a promising new target for the development of therapeutic tools for treating heart failure, and persistent effort in developing and testing *cBIN1* gene therapies will reveal its full therapeutic potential.

## 5. Conclusions

In this review, we provide an overview of the current understanding of t-tubule cBIN1-microdomains and their alterations in heart failure. Cardiomyocyte t-tubule cBIN1-microdomains are a master regulator of intra-myocyte calcium handling and a marker and therapeutic target of heart failure and thus deserve further investigations at the bench and the bedside. Future preclinical studies in large animal models as well as clinical trials in human patients will be needed to further test the biomarker potential of the plasma CS test as well as the therapeutic potential of targeting cBIN1-microdomains in failing cardiomyocytes.

## Figures and Tables

**Figure 1 ijms-22-02299-f001:**
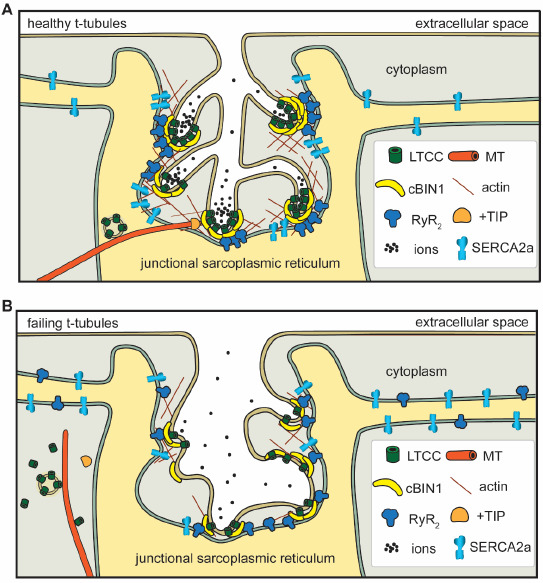
In healthy cardiomyocytes (**A**), t-tubule cBIN1-microdomains regulate intracellular calcium handling through: (1) facilitating forward delivery of L-type calcium channels (LTCCs) via plus-end tracking protein (+TIP) tipped microtubules (MT); (2) clustering LTCCs that are already delivered to t-tubule surface; (3) recruiting ryanodine receptors (RyRs) at the junctional sarcoplasmic reticulum (jSR) for effective dyad formation; (4) organizing a subpopulation of SERCA2a near the jSR; and (5) creating an extracellular slow ion diffusion zone. All of these functions are disrupted in failing cardiomyocytes (**B**).

**Figure 2 ijms-22-02299-f002:**
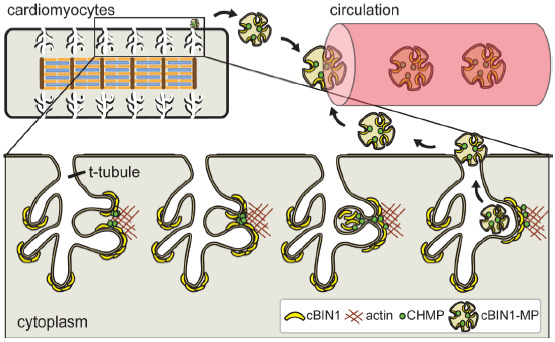
Cardiomyocyte cBIN1-microdomains turn over through CHMP4B-dependent microvesicle formation and release from t-tubule membrane, resulting in cardiac origin cBIN1-vesicles flowing in the peripheral circulation as a marker of cardiomyocyte health.

## Data Availability

Not applicable.

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
