# Peer review of "Cardiac T-Tubule cBIN1-Microdomain, a Diagnostic Marker and Therapeutic Target of Heart Failure"

_ijms, 2021, doi:10.3390/ijms22052299_

Round 1
Reviewer 1 Report
This review summarizes recent studies of the role and utility of Bin1 in the heart. Most of the papers discussed in the review originate from the authors' own lab; a number of reviews have also already been published by the authors on the subject of BIN1. This makes this publication feel somewhat redundant - it is however well written and a concise summary of the recent studies published by the authors, and given that similar studies have not been conducted by other groups it makes it an impossible criticism to address; it also however questions the need for yet another review at this stage. The manuscript includes a detailed section regarding other gene therapy trials which sits oddly and doesn't seem to serve a specific function.
Author Response
We appreciate the comments from the Reviewer. The introduction of other gene therapy trials summarizes the current status and work in the field of gene therapy development for heart failure, indicating an urgent need for new effective therapeutic targets like cBIN1. As suggested by the Reviewer, we have also included additional work in the field with regard to BIN1 isoforms and microdomain regulation in the revised manuscript (please see details in our response to Reviewer 2).
Reviewer 2 Report
Li et al present an interesting and well written review on their more recent work looking at cBIN1 as a biomarker of heart failure progression. Not only do the authors present a convincing outline of the role of cBIN1 markers in detecting difficult to diagnose cardiac etiologies, such as HFpEF, but they also touch upon the potential to use these novel techniques for the treatment of other pathologies, for example, neurological disorders such as Alzheimer’s disease. The authors provide an overview on current heart failure gene therapies, the potential of personalized medicine and what is still needed to be explored on this topic. This review was insightful and enjoyable to read. I have only minor comments that I think will improve the manuscript.
In the basic biology section, the authors nicely describe the role of the different BIN1 exons and isoforms. The authors previous work has identified several newly discovered alternatively spliced variants of BIN1 and one of these variants (cBIN1) is the focus of the review. I think it is still important however that the authors acknowledge that several research groups, including the early work of Hong et al (2010), have demonstrated that other variants of BIN1, especially the muscle variant containing (exon 11) play an important role in the heart. The exon 11 containing variant has been shown to be important for delivery of L-type calcium channels to cardiac t-tubules and for the promotion of t-tubules in cardiac cells (Laury-Kleintop J Cell Biochem (2015); De La Mata, Stem Cells (2019); Lawless et al, Sci Rep (2019)). Furthermore, Lawless et al, Sci Rep (2019) demonstrated that exon 11 containing variant 8 was the most dominantly expressed isoform in the sheep heart. Following on from this point, it would be interesting to hear the authors view point on if the other, more predominately expressed, cardiac BIN1 variants could also be used as a disease biomarker?
Do the authors think that cBIN1 alterations are a cause or consequence of disease and what evidence is there to support this? What about diseases that are not caused by cardiomyocyte pathophysiology, but a combination of other factors?
The section on cBIN1-microdomains was particularly fascinating, I understand there is little known on the topic, but could the authors expand on the benefits of cardiomyocytes releasing cBIN1 to the extracellular environment, for it then to be taken up by another myocyte vs. myocytes making microparticles?
Line 64: @ should be beta
Line 482: cardia should be cardiac
Author Response
We deeply appreciate the support from the Reviewer as well as all the insightful suggestions, which have significantly improved our review manuscript. All the suggested changes have been incorporated in the revised manuscript. Please see below for our point-by-point response:
- In the basic biology section, the authors …… other cardiac BIN1 variants could also be used as a disease biomarker?
Response: We appreciate the careful review from this Reviewer. Our original work prior to the cloning of cBIN1 (Hong et al, PLoS Biology, 2010) used the exon 11 containing skeletal isoform 8 of BIN1 in the cellular studies, which retains the capability in trafficking calcium channels when over-expressed in HL-1 cells. The work from Laury-Kleintop et al (J Cell Biochem, 2015) used the mice line with cardiac specific Bin1 deletion, which removes all BIN1 isoforms including cardiac and skeletal isoforms. Following these studies, both our work (Hong et al, Nature Medicine, 2014) and the work later from other laboratories (unpublished correspondence from colleagues) found that exon 11 containing BIN1 isoforms are not detectable in mouse hearts. However, as you mentioned, a recent study reported that exon 11 containing BIN1 isoforms are expressed in sheep hearts (Lawless et al., Sci Rep, 2019). Similar to HL-1 cells, when over-expressed in human embryonic stem cell‐derived cardiomyocytes, the exon 11 containing BIN1 isoform 8 also contains the capability of trafficking Cav1.2 (De La Mata, Stem Cells, 2019). Thus, the roles of exon 11 containing BIN1 isoforms in cardiac t-tubule biology and calcium handing deserve future more detailed studies. Interestingly, even with the high expression levels of exon 11 containing BIN1 isoforms in skeletal muscle, they are not detectable in both human and mouse plasma. The two isoforms detected in the blood stream are BIN1+17 and BIN1+13+17 (cBIN1). We found that plasma cBIN1 but not BIN1+17 levels correlate with heart failure status of patients. In fact, our original ELISA capturing both isoforms (Hong et al, Heart Rhythms, 2012) does not perform to the same level as the new cBIN1 specific assay for heart failure diagnosis and prognosis. To highlight the results from these studies, in the revised section, we have added the new text as below:
Section 2.2, lines 108-116: “Interestingly, exon 11-containing skeletal BIN1 isoforms absent from mouse cardiomyocytes are expressed in sheep hearts [26], indicating a potential role of skeletal BIN1 in cardiac t-tubule biology. Furthermore, over-expression of the exon 11-containing skeletal BIN1 isoform 8 can induce narrow tubular structures with retained calcium channel trafficking capacity in HL-1 cells [27], human embryonic stem cell‐derived cardiomyocytes [28], neonatal rat ventricular myocytes, and iCell iPSC-derived cardiac myocytes [26]. Future studies will be necessary to understand the expression profiles of BIN1 isoforms and their functional significance during myocyte maturation in humans and other species of large mammal.”
Section 3.2, lines 249-260: “BIN1’s interaction with CHMP4B requires its constitutive N-terminal BAR domain [35], which is present in all BIN1 isoforms. Interestingly, BIN1+17 and cBIN1 are the only two BIN1 isoforms circulating in human and mouse plasma despite the higher expression levels of other BIN1 isoforms in heart muscle. Similarly, the exon 11-contianing BIN1 isoform 8 highly expressed in skeletal muscle is also not detectable in the peripheral blood stream [35]. These data indicate that additional regulation and protein interaction are required for the cellular release of BIN1 containing vesicles. Although the cell origin and functional significance of plasma BIN1+17 vesicles remain unclear, a comparison between the nonspecific assay detecting BIN1+17 and cBIN1 in human plasma [55] and the cBIN1 specific test [58, 59] indicates a lack of correlation between plasma BIN1+17 and cardiac function. Future studies will be necessary to understand the BIN1 isoform-specific regulation of microvesicle release from cells.”
Section 3.3, lines 326-328: “At the time, the utilized non-specific BIN1 blood test captured both BIN1+17 and cBIN1 in human plasma, thus explaining the large variations observed in measured plasma BIN1 concentrations in individuals from the same study cohort [55].”
- Do the authors think that cBIN1 alterations are a cause or consequence of disease …… but a combination of other factors?
Response: We thank the Reviewer for this helpful comment. The transcriptional reduction of cBIN1 can be induced by commons stress responses, such as chronic sympathetic overdrive induced by continuous isoproterenol infusion (Liu et al., JACC BTS, 2020) and pressure overload induced by transverse aortic constriction (Li et al, Front. Physio., 2020). The reduced cBIN1 expression with the resultant disruption of cBIN1-microdomains will then cause the alterations in calcium handling in cardiomyocytes. Given the rescue capability of exogenous cBIN1, we believe that BIN1 alterations are a common cause of some of the most common cardiomyocyte pathophysiology of acquired heart failure. However, future studies are necessary to understand the biology of BIN1 regulation and its contribution to heart failure progression caused by factors outside of cardiomyocyte pathophysiology. In the revised manuscript, we have added the new text as below:
Section 2.3, lines 164-168: “While the cause of this reduction remains unclear, results from recent studies indicate that cBIN1 expression is reduced when the myocardium is subjected to common stressors such as sympathetic overdrive induced by chronic isoproterenol infusion [31] and continuous pressure overload induced by transverse aortic constriction [44].”
Section 2.3, lines 192-199: “This rescue capacity of exogenous cBIN1 further indicates that cBIN1 is a master regulator of the calcium handling machinery in cardiomyocytes and that the disruption of cBIN1-microdomains is a cause of calcium mishandling in failing hearts. The potential of cBIN1 gene therapy for heart failure will be reviewed in detail in Section 4 of the current review. Future studies are also needed to understand whether alterations in intra-myocyte t-tubule cBIN1-microdomains influence heart failure development promoted by factors outside of cardiomyocyte pathophysiology such as inflammation, fibrosis, and remodeling in extracellular matrix.”
- The section on cBIN1-microdomains was particularly fascinating, … for it then be taken up by another myocyte vs. myocyte making microparticles?
Response: We appreciate the great interest from the Reviewer. These are all wonderful thoughts to explore the benefits of cardiomyocytes releasing cBIN1 vesicles to extracellular environment and how the vesicles will impact another myocyte vs. myocyte making microparticles. Indeed, such research is ongoing in our laboratory to further understand the content, destiny, and cellular function of these cardiac origin cBIN1 vesicles. We hope to complete these studies in the near future and share our results with the research community. In the revised review manuscript, we have added new text about the potentials of cBIN1 vesicles. Please see below:
Section 3.2, lines 294-299: “For instance, it is possible that these vesicles may beneficially influence the local extracellular environment of the heart, that these vesicles when reabsorbed may have differential effects on vesicle-releasing versus non-releasing cardiomyocytes, and that the cardiac released vesicles may also travel far in the peripheral circulation to generate a remote impact on other organ systems such as skeletal muscle.”